# Degranulation of Murine Resident Cochlear Mast Cells: A Possible Factor Contributing to Cisplatin-Induced Ototoxicity and Neurotoxicity

**DOI:** 10.3390/ijms24054620

**Published:** 2023-02-27

**Authors:** Betül Karayay, Heidi Olze, Agnieszka J. Szczepek

**Affiliations:** Department of Otorhinolaryngology, Head and Neck Surgery, Charité-Universitätsmedizin Berlin, Corporate Member of Freie Universität Berlin, Humboldt-Universität zu Berlin, 10117 Berlin, Germany

**Keywords:** ototoxicity, cisplatin, mast cells, cromoglicic acid, cromolyn, cochlea, mouse

## Abstract

Permanent hearing loss is one of cisplatin’s adverse effects, affecting 30–60% of cancer patients treated with that drug. Our research group recently identified resident mast cells in rodents’ cochleae and observed that the number of mast cells changed upon adding cisplatin to cochlear explants. Here, we followed that observation and found that the murine cochlear mast cells degranulate in response to cisplatin and that the mast cell stabilizer cromoglicic acid (cromolyn) inhibits this process. Additionally, cromolyn significantly prevented cisplatin-induced loss of auditory hair cells and spiral ganglion neurons. Our study provides the first evidence for the possible mast cell participation in cisplatin-induced damage to the inner ear.

## 1. Introduction

Mast cells (MCs) attract the attention of numerous research disciplines, but in one of them—inner ear biology—the presence and role of MCs were scarcely investigated. The inner ear is a sensory organ responsible for auditory and vestibular perception. Well-secluded in the temporal bone and protected by a blood–labyrinthine barrier, the inner ear is guarded against the external and internal environment. For decades, the clinical and basic research in inner ear biology focused on the sensory epithelium and the auditory or vestibular neurons. Recently, the role of innate and acquired immunity started to be recognized, and many types of resident immune cells were identified in the inner ear [1,2]. Concerning the MCs, Poulsen (1959) was the first to describe their presence in the upper semicircular canal of the guinea pig [3]. Two decades later, Sleeckx et al. (1979) identified MCs in the human inner ear’s endolymphatic sac [4]. Finally, functional studies using the guinea pig model of type-I allergy demonstrated allergen-induced MC degranulation in the endolymphatic sac concurrent with nystagmus [5,6] and temporary hearing loss. These symptoms could be prevented by an inhibitor of MC histamine release, tranilast [7], or MC stabilizer pemirolast [8]. Finally, our earlier study described the resident MCs in the cochlea of mice and rats [9].

Hearing loss affects 5% of people worldwide and is the most common communication disorder [10]. Clinical evidence links mast cell disorders to inner ear illnesses. In patients with congenital or familial mastocytosis, sensorineural (cochlear) hearing loss was diagnosed [11,12]. Furthermore, sensorineural hearing loss is common among patients with allergic rhinitis, and MC involvement was suggested as a possible mechanism [12]. Finally, in a patient with cutaneous mastocytosis and Meniere disease, therapy with omalizumab relieved the mastocytosis symptoms and those caused by Meniere disease affecting the inner ear [13].

Hearing loss can often be caused by ototoxic medications [14]. One such medication inducing irreversible damage to the inner ear is cisplatin, commonly used for cancer therapies [15]. The onset of hearing loss begins days to months after the treatment and initially impacts only the high-frequency range but can progress to the low-frequency range [16], leaving patients with a permanent communication disorder.

Upon entering a cell, cisplatin binds the DNA and forms so-called DNA adducts, which can activate various signaling pathways. That activation may induce DNA repair, cell survival, or cell death [17]. Cancer cells frequently have disrupted DNA damage repair machinery and die, whereas non-neoplastic cells repair their DNA and stay alive. The first reason for the inner ear’s vulnerability to cisplatin is the expression of platin uptake receptors such as organic cation receptor 2 (OCT2) present on hair cells and the stria vascularis (SV) [18] or the copper transporter (CTR1) expressed on outer and inner hair cells (OHCs and IHCs), SV, and spiral ganglion neurons (SGNs) [19]. Another critical reason for cisplatin-induced ototoxicity is the absence of cisplatin clearance from the cochlea and long-time retention observed in the mouse model and human temporal bone specimens [20]. In addition, cisplatin targets Deiters’ and Hensen’s cells, which are essential for supporting OHC function [21]. Lastly, cisplatin induces the loss and degeneration of the mitochondria in SGNs [22], causing hearing loss [23]. The molecular pathomechanism of cisplatin ototoxicity is associated with the production and accumulation of reactive oxygen species (ROS) by the hair cells, leading to their death [24,25]. The ototoxicity of cisplatin overlaps with its neurotoxic properties [26,27], causing a loss of auditory SGNs and further contributing to hearing loss [28]. Despite significant progress in ototoxicity research, the molecular mechanisms responsible for the cochlear structures’ degeneration in cisplatin’s presence are still the focus of scientific exploration.

In our earlier study, we demonstrated the presence of MCs in the cochlea of rodents [9] and observed that cisplatin modulates their numbers. However, the resident cochlear MCs’ role in the ototoxic impact of cisplatin has not yet been addressed. Therefore, the present study aimed to elucidate the effects of cisplatin on the murine cochlear MCs and determine whether inhibiting MC degranulation using cromolyn could protect the hair cell and SGN damage in cochlea exposed to cisplatin.

## 2. Results

All experiments were performed using explanted cochlear membranous tissues dissected from P3-P5 C57Bl/6 mice pups. The MCs were identified with avidin–Alexa Fluor ™ 488, the hair cells with phalloidin-iFluor 594, and the SGNs with an antibody against neurofilament-200. The phalloidin-stained hair cells were scrutinized and scored under the fluorescent microscope to assess the degree of cisplatin-induced hair cell damage. Only the cells with a non-disrupted stereocilia array were counted as “intact”. For detailed methods, see Section 4.

### 2.1. Cisplatin Affects the Numbers and Morphology of Cochlear Mast Cells

Earlier studies demonstrated that cisplatin activates peritoneal MCs to release histamine [29]. To confirm if the cochlear MCs also degranulate in the presence of cisplatin, we analyzed the distribution of MCs in the cochlear membranes, determined if the cisplatin-mediated degranulation takes place, and examined the numbers and morphology of MCs after exposure to different concentrations of cisplatin.

#### 2.1.1. Distribution of Cochlear Mast Cells in the Cochlear Explants

Each cochlear explant consisted of a spiral limbus containing the organ of Corti (OC) with the auditory hair cells and SGN. The cochlear lateral wall (LW) explants contained the SV and spiral ligament (SL).

To investigate the distribution of cochlear MCs in the membranous cochlea, we divided the tissue into apical, medial, and basal parts representing the low, middle, and high hearing frequencies. Significantly more resident MCs were present in the apical than in the medial or basal part of the cochlea (Figure 1b). Representative images of the MCs in the cochlear explants are shown in Figure 1c–f.

Cochlear MCs were found in the spiral limbus proximal to SGNs (neurites and cell bodies), while no MCs could be observed directly in the OC (Figure 1a). There were on average more MCs in the LW than in the spiral limbus (Figure 1a). The mean total number of MCs was 0.37 ± 0.06 in the spiral limbus and 1.85 ± 0.53 in the LW.

#### 2.1.2. Cisplatin Leads to Degranulation of Cochlear Mast Cells

The cochlear explants were cultured either in the tissue culture media only or exposed to 10 µM, 15 µM, 20 µM, and 30 µM cisplatin for 24 h and then stained with avidin–Alexa Fluor 488 conjugate. Representative images of the MCs in the LW of the respective treatment groups are shown in Figure 2a–e. The control group (Figure 2a) contained intact MCs with cytoplasmic granules labeled by avidin. The granules were frequently scattered into the extracellular space, consistent with MC degranulation in the cisplatin-exposed groups (Figure 2c–e).

#### 2.1.3. Cisplatin Decreases the Number of Non-Degranulated Cochlear Mast Cells

The number of non-degranulated MCs decreased after exposure to cisplatin (Figure 3a). On average, there were 1.89 ± 0.76 of non-degranulated MCs in the untreated explants and 1.23 ± 0.33 of degranulated MCs. After exposure to 15 µM, 20 µM, and 30 µM cisplatin, the average number of non-degranulated MCs decreased to 0.25 ± 0.09, 0.70 ± 0.24, and 0.06 ± 0.06, respectively. Exposure to 10 μM cisplatin reduced the numbers of the non-degranulated MCs, but this reduction was not statistically significant (Figure 3a). A significant difference in the average number of degranulated MCs was not seen in any treatment groups (Figure 3b). An increase in degranulated MCs was only observed after exposure to 10 µM cisplatin (Figure 3b). Moreover, as demonstrated in Figure 3c, the number of MCs in the explants decreased with increasing cisplatin concentration (OC and LW combined). On average, there were 1.58 ± 0.42 MCs in an untreated explant, regardless of the MC degranulation stage. After exposure to 15 µM cisplatin, the average number of MCs decreased, but the decrease was significant only in the 20 µM and 30 µM groups. In the group exposed to 10 µM cisplatin, an increase in MCs to 2.31 ± 0.75 was recorded; however, that increase was not statistically significant.

### 2.2. Cromolyn Protects the Auditory Hair Cells from Cisplatin-Induced Ototoxicity

After determining that exposure to cisplatin leads to a decrease of non-degranulated cochlear MCs, we wanted to examine if an inhibition of MC degranulation correlates with cisplatin-induced hair cell damage and loss. Therefore, we used a commercially available, clinically and experimentally approved MC stabilizer, cromolyn.

#### 2.2.1. Determination of Biosafety of Cromolyn in the Cochlear Tissues

To determine if cromolyn alone affects the hair cells, cochlear explants were cultured for 24 h in the presence of cromolyn (5 µM, 10 µM, 25 µM, 50 µM, 100 µM, and 200 µM). The tissues were fixed and stained with phalloidin to visualize the hair cells. Cromolyn used at the concentrations of 100 µM and 200 µM led to a significant decrease in intact inner hair cells (IHCs) and outer hair cells (OHCs) compared to the control group (Figure 4a). Representative images of the hair cells from the respective treatment groups are shown in Figure 4d–i. The control group (Figure 4b) and the tissues treated with 5 µM, 10 µM, 25 µM, and 50 µM cromolyn (Figure 4d–g) contained intact IHCs and OHCs. The hair cells were damaged and lost their normal morphology in the group exposed to 100 µM and 200 µM (Figure 4h,i).

#### 2.2.2. Determination of the LD_50_ for Cisplatin

The cochlear explants were exposed to cisplatin (5 μM, 10 μM, 15 μM, and 20 μM) for 24 h (Figure 5). The number of intact hair cells decreased with the increasing concentration of cisplatin. Cisplatin concentration inducing damage to 50% of the IHCs (Figure 5a) and OHCs (Figure 5b) was used as the LD_50_. Based on the LD_50_ for the OHCs (12.43 µM) and LD_50_ for IHCs (13.54 µM), the cisplatin LD_50_ concentration was arbitrarily set at 15 µM for all remaining experiments.

#### 2.2.3. Cromolyn Inhibits the Cisplatin-Induced Degranulation of Cochlear Mast Cells

The cochlear explants were exposed for 24 h to 15 µM cisplatin and various concentrations of cromolyn. Statistical analysis indicated that 5, 10, and 25 µM cromolyn efficiently blocked the MC degranulation process in the cochlea (Figure 6a). Cromolyn used at 50 µM was no longer effective in blocking cisplatin-induced MC degranulation. Representative images of the MCs in the LW of the respective treatment groups are shown in Figure 6b–g. The control group (Figure 6b) and the group pretreated with 5 µM, 10 µM, and 25 µM cromolyn (Figure 6d–f) contained intact MCs with cytoplasmic granules labeled by avidin. The cisplatin-exposed tissues (Figure 6c) and the group pretreated with 50 µM cromolyn and exposed to cisplatin (Figure 6g) contained avidin-positive granules scattered into the extracellular space.

#### 2.2.4. Cromolyn Protects Auditory Hair Cells from Cisplatin-Induced Damage

In contrast to the control explants (Figure 7a) and explants treated with cromolyn only (Figure 7b,c), we observed hair cell damage in the cisplatin-exposed explants (Figure 7d). Analysis of cochlear explants pretreated with cromolyn for 2 h and then exposed to 15 μM cisplatin for 24 h demonstrated significant protection of hair cells by cromolyn (Figure 7e,f). The cochleograms representing the mean values of intact IHCs and OHCs are shown below each micrograph.

To investigate the extent to which cromolyn protects against cisplatin-induced hair cell damage and loss, the intact hair cells were counted, and the statistical significance of differences between the treatment groups was calculated (Figure 8). Cisplatin has induced significant damage and loss of IHCs (Figure 8a) and OHCs (Figure 8b). The damage and loss of IHCs were significantly reduced by preincubation with 5 and 25 µM of cromolyn. In contrast, the damage and loss of OHCs were reduced considerably by preincubation with 5 µM but not 25 µM of cromolyn.

### 2.3. Cromolyn Protects from Cisplatin-Induced Neurotoxicity

We also investigated whether cromolyn influences the neurotoxic effect of cisplatin. The cochlear explants (spiral ligament with OC and SGN) were either cultured for 24 h with various concentrations of cromolyn (5 µM, 10 µM, 25 µM, and 50 µM) or pretreated for 2 h with cromolyn (5 µM, 10 µM, 25 µM, and 50 µM) and then exposed to 15 µM cisplatin for a further 24 h. The region of interest (ROI) area containing SGN between the habenula perforata and IHCs was calculated in a binary image and compared to the control groups (Figure 9a). Cisplatin led to a significant decrease in SGN compared to the control group. Pretreatment with 10 µM of cromolyn alone significantly decreased the SGN numbers. No significant differences were found when using other cromolyn concentrations. SGN protection was significant when the specimens were pretreated with 25 µM cromolyn before exposure to cisplatin (Figure 9a). Representative images of the SGNs and the associated binary images of the respective treatment groups shown in Figure 9b–g were used to evaluate the SGN density.

## 3. Discussion

This work aimed to investigate the involvement of cochlear MCs in the pathological processes induced by cisplatin in the cochlea. Cisplatin caused the degranulation of murine cochlear MCs, and cromolyn inhibited the cisplatin-induced MCs’ degranulation. Moreover, adding cromolyn to cisplatin-exposed cochlear tissues correlated with a decreased loss of auditory hair cells and SGN.

### 3.1. Brief Overview of Cisplatin-Induced Adverse Effects and Their Mechanisms

Several mechanisms underlying cisplatin-induced ototoxicity have been proposed to date. Direct mechanisms involve active intake [18,19] and the accumulation and retention of cisplatin in cochlear tissues [20]. After transport to the cochlea, cisplatin binds DNA, forming adducts and activating various pathways. Activation of pathways involving ATR, p53, p73, and MAPK leads to cell death, whereas activation of the PI3-K/Akt pathway leads to cell survival [17]. The involvement of p53 in cisplatin-induced ototoxicity has been studied, and inhibition of p53 has been demonstrated to be otoprotective [30]. Similarly, inhibiting the mitogen-activated protein kinase MAPK induced by cisplatin protected the auditory hair cells from death [31]. Moreover, activation of the PI3-K/Akt pathway protects from cisplatin-induced ototoxic damage [32,33]. For a detailed analysis of contemporary knowledge about ototoxic and protective mechanisms, see the respective reviews [34,35,36,37]

The DNA adducts impair the transcription and translation of detox enzymes [38,39]. Since cisplatin induces ROS production, causing oxidative stress [25], cisplatin-initiated ROS accumulation and the antioxidant system’s shutdown ends in cell damage and apoptosis [40,41,42]. In addition, cisplatin-induced inflammation and consecutive secretion of proinflammatory cytokines contribute to cochlear damage, even before the production of ROS [43]. However, no particular cell type producing the cytokines in the cochlear tissues in response to cisplatin was yet identified.

Ototoxicity is only one of many adverse effects of cisplatin, including nephrotoxicity, cardiotoxicity, and hypersensitivity reactions [44]. Nephrotoxicity, similar to ototoxicity, is primarily attributed to a high cisplatin uptake by proximal tubule cells expressing the transporters CTR1 and OCT2 (reviewed in [36]). The common denominator for the above side effects appears to be the MC and its mediators released in the presence of cisplatin. In agreement with that, bradycardia induced by cisplatin was implicated to be a consequence of a hypersensitivity reaction characterized by a release of histamine, prostaglandins, and other MC mediators [45]. Moreover, the nephrotoxic responses to cisplatin were connected to TNF-alpha released by MCs during acute kidney injury induced by cisplatin [46]. Lastly, MCs will most likely mediate hypersensitivity type I responses to cisplatin treatment [47]. Taken together, the axis mast cell–cisplatin-induced pathology seems to be a promising topic for translational studies focusing on preventing cisplatin’s adverse effects.

### 3.2. The Effect of Cisplatin on Murine Cochlear Mast Cells

In the present work, we asked if cisplatin could induce the degranulation of cochlear MCs. Indeed, we observed a significant reduction of non-degranulated MCs in the explanted limbal and lateral wall tissues, indicating that cisplatin induces either death or degranulation of cochlear MCs. Our results corroborate those of Brzezińska-Błaszczyk et al. (1996), who showed that peritoneal MCs release histamine in the presence of cisplatin [29]. Since histamine is found in the MC granules [48], its release is consistent with degranulation. We also demonstrated that exposure to 20 µM and 30 µM of cisplatin leads to a significant decrease in the total number of MCs, indicating a dose-dependent effect either resulting in degranulation of MCs or inducing apoptosis. In agreement with that, experiments with the human MC line HMC-1 demonstrated that high cisplatin concentrations induce MCs’ apoptosis [49]. It can be concluded that the effect of cisplatin on MCs is either degranulation or apoptosis, depending on the concentration. Completely degranulated MCs are no longer detected by the staining method with avidin. Although the use of the glycoprotein avidin is an established staining method for identifying MC granules [50], after degranulation, the granules are scattered in the extracellular space, which may result in some MCs remaining undetected and thus neglected during quantification.

Our earlier work has shown that the exposure of cochlear explants to 20 µM cisplatin paradoxically increases the total number of cochlear MCs in Wistar rats [9]. In the present work, upon exposure to 20 µM cisplatin, a significant decrease in the number of MCs could be observed. One possible reason for this discrepancy is using different animal species and a distinct postnatal developmental stage—in the previous work, 5-day-old Wistar rats were used, while here we used 3–5-day-old C57BL/6 mice. The observation of increased MC numbers upon exposure to cisplatin is supported by Takagi et al., who also observed an increased number of MCs after activation with IgE and attributed it to the presence of undifferentiated MCs, which began to differentiate after stimulation [51]. The proliferation and differentiation of the MCs occur in the presence of stem cell factor (SCF), which can be produced by resident cochlear macrophages [52] and, similarly to TNF-alpha and IL-1 beta [53], released upon exposure to cisplatin. Future studies should expand this observation.

During the experiments investigating the effect of cisplatin titration on the degranulation of cochlear MCs, when using 15 µM cisplatin, we observed a significant decrease in intact MCs that was not compensated by an increase in degranulated MCs. This decrease was more prominent than when using 20 or 30 µM cisplatin. A possible explanation of this result is that cisplatin exhibits pleiotropic effects that depend on a cell type. This work used primary cochlear tissues composed of sensory, neuronal, epithelial, fibrocyte, and resident immune cells. It is possible that exposing the cochlear tissues to various concentrations of cisplatin might result in multiple composite cellular and acellular (e.g., via cytokines [54]) responses, indirectly affecting the number or degranulation status of MCs. Additional studies using a purified population of MCs should clarify if the effect of 15 µM cisplatin on MC degranulation was due to direct or indirect interactions.

The last possible mechanism in which cisplatin-mediated degranulation of MC could cause cochlear damage is proteolytic damage to the cochlear tissues and interference. MC granules contain several proteases [55] that can target tight junctions disrupting epithelial or connective tissue barriers [56,57]. Since connexins and claudins representing gap junction proteins are indispensable in the inner ear for potassium recycling and maintaining endocochlear potential, loss or damage of junction proteins could result in hearing loss, as demonstrated by hereditary human deafness associated with mutations affecting connexin 26, 29, 30, 31, and 43 [58]. However, in the present work, the issue of tight junctions and MC proteases was not studied because of the model we used. It should, however, be addressed in the future using an animal model.

### 3.3. Effect of Cromolyn on Cisplatin-Induced Cochlear Mast Cell Degranulation

Upon adding 5, 10, or 25 µM of cromolyn to cochlear explants and exposing them to cisplatin, we observed significant inhibition of MC degranulation, as per avidin staining. In a study by Oka et al. [59], cromolyn used at concentrations between 10 and 100 µM failed to inhibit the IgE-induced activation of mouse- but not rat-derived MCs as measured by β-hexosaminidase release. However, cromolyn was added simultaneously with the stimulant in these experiments, possibly accounting for this difference. In addition, we used cisplatin and not IgE to stimulate MCs, and although the exact mechanism of MC stimulation by cisplatin remains unknown, it is unlikely identical to that of IgE. Lastly, we observed the degranulation microscopically rather than measuring mediators in the supernatant. In agreement with our findings, other groups reported successful in vitro use of cromolyn to block murine MC degranulation induced by the vaccinia virus [60] or in the mouse model of vasopressor responses to the vasoactive peptide endothelin-1 [61].

### 3.4. Effect of Cromolyn on Cisplatin-Induced Hair Cell Damage

Cromolyn significantly inhibited cisplatin-induced auditory hair cell loss when used at 5 or 25 µM for a 2 h pretreatment before adding cisplatin to the cochlear explants. No base–apex gradient was observed for the protective effect of cromolyn. The cellular cochleograms obtained after in vitro exposure of cochlear explants to cisplatin agreed with the observations made by Ding et al., who used a similar in vitro experimental system [62].

This new observation could likely reflect a known mechanism of cisplatin-induced toxicity described in the kidney. The resident MCs were suggested to contribute to the cisplatin-induced acute kidney injury (AKI) via TNF-alpha release [46]. Inhibition of MC degranulation with cromolyn significantly reduced AKI.

Another possible mechanism, which has not been studied here but should be discussed, is the effect of cromolyn on resident macrophages and microglia. In a recent study, Wang et al. determined the impact of cromolyn on resident macrophages in the central nervous system (microglia) [63]. In that study, a human microglial cell line HMC3 was stimulated with TNF-alpha to release various cytokines and chemokines, and that secretion was inhibited by cromolyn. The microglia-like cells are present in the cochleae of p5 mice [64]. In addition, perivascular-resident macrophage-like melanocytes residing in the stria vascularis were shown to secrete IL-1 beta in response to cisplatin, contributing to cisplatin-induced cochlear inflammation [65]. Inflammation and secretion of proinflammatory cytokines are considered the first stage of cisplatin-induced cochlear damage, even before the production of reactive oxygen species [66]. However, although the cytokine expression in response to cisplatin was determined, no particular cell type producing the cytokines in the cochlear tissues has yet been identified.

The last known mechanism of cromolyn’s action is decreased reactive oxygen species production. In alveolar macrophages, cromolyn decreased ROS production, preventing ROS-induced lung damage [67]. Cromolyn-dependent inhibition of cisplatin-induced ROS production in the cochlea could likely reduce its ototoxic effect.

### 3.5. Effect of Cromolyn on Cisplatin-Induced Spiral Ganglion Neuron Loss

In the last part of our work, we demonstrated that cromolyn protects spiral ganglion neurons from cisplatin-induced toxicity. Cisplatin can be neurotoxic [26,27] and reduce the number of SGNs, contributing to hearing loss [28]. In agreement with the physiological data, cisplatin was demonstrated to induce cell death of cultured sensory neurons from rats in a dose-dependent fashion [68]. Here, we showed that cromolyn inhibits the cisplatin-induced SGN loss in the cochlear explants but only at the concentration of 25 µM. The reason why only this particular concentration of cromolyn was effective remains unclear and needs to be investigated further. Previously described neuroprotective abilities of cromolyn were associated with the inhibition of cerebral MCs, leading to improved cognitive function after hypoxic–ischemic brain damage [69]. Kempuraj et al. (2020) showed that treating mice with cromolyn protects against brain damage induced by acute trauma [70]. In addition to stabilizing action on MCs, the inhibition of proinflammatory cytokines secretion from microglia [45] also plays a critical role here.

During the experiments where various cromolyn concentrations were used to determine possible effects on the SGNs, we observed a significant drop in the neurite density when using 10 µM but not lower (5 µM) or higher (25 or 50 µM) cromolyn concentrations, suggesting a nonlinear response to cromolyn. Cromolyn binds to proteins from the S100 family [71], expressed in the developing cochlear tissues in mice on the hair cells, supporting cells, and spiral limbus [72]. Since the Ca^2+^-binding S100 can regulate a wealth of physiological intracellular and extracellular processes, including neurotrophy [73], it is plausible that cromolyn at a particular concentration might interfere with that process in the still-developing cochlea. Another target of cromolyn is the G-protein coupled receptor 35 (GPR35). Recent studies suggested that cromoglicic acid acts as a selective agonist of GPR35, modulating the intracellular Ca^2+^ release [74,75]. The expression pattern of GPR35 in the cochlea is unknown, but it is known that various immune cells, including MCs, express the GPR35 on their surface [74,76]. Here, the nonlinear type of response to cromolyn is also puzzling and requires further investigation.

### 3.6. Study Strengths and Limitations

The first major limitation of this study is the experimental system used. The auditory system of P3-P5 mice used in the present work is still immature, as the hearing onset in mice occurs after P12 [77,78]. Therefore, the cochlear tissues explanted from the young animals will differ physiologically from fully mature but no longer explantable tissue. Still, based on the method described by Hanna Sobkowicz in 1975 [79], tissue culture of the membranous cochlear tissue from mice has been used to study in vitro the ototoxicity of various drugs [31,41,80,81]. This system allows studying the intact morphology of the auditory periphery, which is a strength of our study. It is, however, still possible that the fully mature hair cells and spiral ganglion neurons could react differently. Nevertheless, such preliminary screening studies warrant further research using the animal model according to animal welfare policies.

A different pitfall of our study is a high degree of variability regarding the numbers and distribution of the cochlear MCs. More MCs were found in the lateral wall than the spiral limbal tissues. On average, one MC could be identified in the spiral limbus (containing OC, SGN, and SG cell bodies), whereas two MCs were present in the lateral wall. Notably, the fully degranulated MCs (e.g., during the tissue dissection and preparation process) cannot be visualized with avidin, contributing to the specimen variation. The glycoprotein avidin conjugated with the fluorescent dye Alexa FluorTM is frequently used to identify MCs and is an established staining method during fluorescent imaging of mast cells [50]. The degranulation of MCs can either be initiated via ligand binding to the high-affinity receptor for IgE (FcεRI), by activation of other receptors [82,83,84,85,86,87] or low temperature [88,89]. Since cochlear explants are prepared at 4 °C, one possible reason for the varying numbers of avidin-positive MCs observed in the cochlear tissues could be the methodological approach, leading to early MC degranulation. However, the MCs would likely re-granulate during further incubation for 24 h at 37 °C. Still, working with native cochlear MCs is seen as a strength of this study. An additional problem, which has not been addressed in the current work, was the possible death of mast cells, its type, and the routes leading to it. These types of experiments require greater numbers of MCs and other detection methods (for example, flow cytometry) coupled with a time curve and are planned to be performed in the future.

Another drawback of this work is the unclear physiological role of cochlear MCs. A part of the lateral wall—stria vascularis—comprises a capillary network composed of three cell layers: marginal, intermediate, and basal cells [90]. The intermediate cell layer consists of the perivascular resident macrophages (PVMs) responsible for the homeostasis of the blood–labyrinth barrier (BLB) [91,92]. It is tempting to speculate that MCs could have a modulating impact on the permeability of BLB. This notion is supported by MC-producing mediators such as histamine that have a vasodilating effect and could increase capillary permeability [93]. In the spiral limbus, MCs were located close to the SGN fibers and cell bodies but on no occasion in direct proximity to the IHCs and OHCs. These observations agree with the general knowledge about the MCs’ location [94], permitting neuroimmune interactions mediated by the MCs-derived histamine or serotonin. In the cochlea, the serotonin receptor (5-HTR) [95] and all four histamine receptors [96] are expressed in a variety of cell types, but their physiological role in the cochlea is not yet well known. MCs could also be a source of the nerve growth factor (NGF) necessary for developing SGN [97,98]. Further studies should confirm the physiological significance of cochlear MCs.

### 3.7. Future Directions

The present work demonstrated that cromolyn could significantly prevent the cisplatin-induced MC degranulation and the oto- and neurotoxicity in the murine cochlear explants. This finding implies cochlear MCs’ possible involvement in cisplatin-induced cochlear toxicity. Further in vitro and in vivo preclinical experiments need to verify the obtained results. Future experiments using MCs purified from the peritoneum or bone marrow of experimental animals would be of special importance. Such an approach would enable precise measurement of the released mediators and allow for performing the time course experiments. Another extension of the present work would be a clinical trial in which patients receive cromoglicic acid before cisplatin treatment, and their hearing is subsequently assessed during therapy.

Furthermore, the actual mechanism in which MCs contribute to cisplatin-induced ototoxicity needs to be explored by investigating the released mediators and the biochemical and molecular processes induced in the cochlea by MC degranulation. That could result in promising therapeutic approaches for treating cisplatin-induced hearing loss. Additionally, an extended collaboration between otorhinolaryngologists, immunologists, and dermatologists is advocated to address further questions and improve patient welfare.

Lastly, particular attention should be paid in the future to the prevention of mitochondrial dysregulation, which has not been studied here. Such dysregulation has been described as the primary mechanism in the pathogenesis of cisplatin-induced toxicity [99]. In addition, cisplatin can accumulate in the mitochondrial matrix and negatively affect the antioxidant system, decreasing intracellular ATP and inducing ROS production, which causes mitochondrial dysfunction and activates apoptotic pathways [25,40,41,42]. Interestingly, it has been shown that the levels of peroxisome proliferator-activated receptor gamma coactivator 1-alpha (PGC-1α), a protein that regulates mitochondrial biogenesis [100], decrease during this process [101]. Polyphenols, the largest group of phytochemicals, were demonstrated to regulate mitochondrial biogenesis, positively influence mitochondrial processes, and have anti-neurodegenerative properties [102,103]. Therefore, it is tempting to suggest using polyphenols (e.g., curcumin or resveratrol) during chemotherapy with cisplatin to protect the auditory periphery from the sensory neurodegenerative effects of cisplatin, particularly because these agents can synergize in the potentiating anti-tumor effect of other medications [104]. Since polyphenols can decrease MC activation and degranulation [105], they may offer natural dietary protection from various pathogenic mechanisms induced by cisplatin in the inner ear.

## 4. Materials and Methods

The experimental protocol was approved by the Governmental Ethics Commission for Animal Welfare (LaGeso Berlin, Germany; approval numbers: T 0292/16 granted on 7 December 2016). Newborn C57BL/6 mice were purchased from the local animal facility of the Charité-Universitätsmedizin Berlin. The animals were of both genders and were 3 to 5 days old.

### 4.1. Cochlear Explants

The explants were prepared as previously described [9]. Briefly, after the decapitation of p3-p5 mice, the cochleae were dissected from the temporal bone and placed under the stereoscope SteREO Discovery, V8 (Carl Zeiss, Jena, Germany). After removing cartilage and the bony capsule, the LW, containing stria vascularis and the spiral ligament, was separated from the part containing modiolus and spiral limbus with SGN and the OC. Next, the tissues were divided into apical, medial, and basal parts and explanted in the 4-well culture dishes containing 500 µL of tissue culture medium (DMEM/F12; cat. # 21331-020, Gibco^®^, Carlsbad, Germany), supplemented with 10% heat-inactivated fetal bovine serum (FBS, cat. # S0113, Biochrom AG, Berlin, Germany), 2.5 M glucose (cat. # G8769, Sigma Aldrich, Darmstadt, Germany), insulin-transferrin–Na-selenite (cat. # 11207500, Roche, Basel, Switzerland), penicillin G (cat. # A321-42, Biochrom AG, Berlin, Germany), and IGF-1 (cat. # 4326, RG R&D Systems, Wiesbaden-Nordenstadt, Germany). The culture was conducted in a humidified incubator at 37 °C and 5% CO_2_ for 24 h. The explants were fixed in 10% formalin (cat. # HT5011, Sigma-Aldrich, Darmstadt, Germany) for 40 min at room temperature (RT), rinsed with PBS, and kept in PBS at 4 °C before immunofluorescence staining.

### 4.2. Treatments of the Explanted Tissues

All substances were freshly prepared for each experiment under sterile conditions. The control groups assigned to each treatment were incubated with culture media only. Each type of experiment was repeated two to four times.

#### 4.2.1. Treatment with Cisplatin

Cisplatin (# 232120, Sigma-Aldrich, Darmstadt, Germany) was dissolved in dimethyl sulfoxide (DMSO) to a concentration of 100 mg/mL. This solution was diluted 1:100 in RPMI 1640 medium to the final stock solution (3.3 mM), aliquoted, and stored at −80 °C. The mean lethal concentration (LC_50_) of cisplatin was determined by exposing the explants to cisplatin at the following concentrations: 5 μM, 10 μM, 15 μM, and 20 μM for 24 h. The test explants were compared to the control cultured under the same conditions with the sole culture medium.

#### 4.2.2. Treatment with Cromoglicic Acid

A stock solution of cromolyn (cat. # T1260, Sigma-Aldrich, Darmstadt, Germany) was prepared by dissolving cromolyn in distilled water to a concentration of 50 mM. The possible toxicity of cromolyn was tested by incubating the explants with 5 µM, 10 µM, 25 µM, 50 µM, 100 µM, and 200 µM of cromolyn for 24 h. In the experiments with cisplatin exposure, the explants were preincubated with 5 μM, 10 μM, 25 μM, or 50 μM cromolyn for 2 h and then rinsed twice with DMEM/F12 for 5 min and then exposed to 15 μM cisplatin for 24 h.

### 4.3. Immunofluorescent Staining

#### 4.3.1. Hair Cell Staining

The explants were permeabilized with 0.5% Triton X-100/PBS (# 9002-93-1, Sigma Aldrich, Darmstadt, Germany) and then incubated with phalloidin-iFluor 594 (cat.# ab176757, Abcam, Cambridge, UK) diluted 1:1500 in PBS. Phalloidin specifically binds the filamentous actin, visualizing the actin-rich stereocilia of the hair cells under the fluorescence microscope. The IHCs and OHCs can be distinguished by their typical morphological arrangement in the OC.

#### 4.3.2. SGN Staining

After permeabilization of the explanted tissues with 0.5% Triton X-100 (cat.# 9002-93-1, Sigma Aldrich, Germany) in PBS, the specimens were incubated with 4% blocking solution (goat serum cat.# 005-000-121, Jackson ImmunoResearch Europe Ltd., Ely, UK) for 60 min at room temperature (RT) to block non-specific binding sites. The explants were then incubated with the primary antibody, mouse monoclonal anti-neurofilament-200 (cat. # N0142, Sigma-Aldrich, Darmstadt, Germany, dilution 1:400) for 40 min at RT and then rinsed with PBS. The secondary antibody was the goat anti-mouse IgG (H + L) Cross-Adsorbed Secondary Antibody, Alexa Fluor 488 (cat. # A 11001, Thermo Fisher Scientific, Karlsruhe, Germany; dilution 1:400) prepared in Antibody Diluent Solution (cat. # ab64211, Cell Signaling Technology Europe, Frankfurt am Main, Germany). The specimens were incubated with the antibodies for 60 min at RT.

#### 4.3.3. Mast Cell Staining

The specimens were permeabilized as described above and incubated with avidin conjugated with Alexa Fluor 488 (cat. # A21370, Thermo Fisher Scientific, Karlsruhe, Germany; dilution 1:600). The nuclei were labeled with ProLong™ Gold Antifade Mountant with DAPI (cat. # P36931, Thermo Fisher Scientific, Karlsruhe, Germany), which fixed the coverslip in place and prevented fluorescence fading.

#### 4.3.4. Fluorescence Microscopy

Digital imaging was performed using an epifluorescence microscope EVOS FL Cell Imaging System (Thermo Fisher Scientific, Karlsruhe, Germany) and the Keyence BZ-X800 (Keyence, Osaka, JPN) microscope with ×10, ×20, and ×40 objectives. The Alexa-488, Alexa-594, and DAPI were excited using argon laser (488 nm), helium-neon laser (543 nm), and blue diode laser (405 nm), respectively. The images were pseudo-colored with RGB tools and reconstructed with ImageJ 1.52q software (http://rsb.info.nih.gov/ij/ (accessed on 12 October 2020)). The quantitative image analyses were performed using ImageJ 1.52q software (https://imagej.nih.gov/ (accessed on 12 October 2020)).

### 4.4. Data quantification

#### 4.4.1. Hair Cell Counting

The cochleogram was created for each cochlear explant to identify the hair cell loss or damage in the respective control and treatment groups. A cochleogram shows the percentage of intact hair cells as a function of distance from the base to the apex of the cochlea. The images of the entire basilar membrane (containing the OC, spiral limbus, and SGNs) were recorded under the 10× objective of the epifluorescence microscope. The ImageJ (Version 1.52q, Wayne Rasband, U. S. National Institutes of Health, Bethesda, MD, USA) plugin Measure_Line was used to estimate the basilar membrane’s length between the lamina spiralis ossea and the IHCs. Next, the specimens were divided into ten sections, each representing a specific part of the total OC length (0% = apex − 100% = base, see Figure 10). Images of three random areas in each section were captured with a 40× objective. Since the IHCs and OHCs were not in the same plane, several images were obtained on different plane levels. These images were then merged into a Z-stack using ImageJ. The IHCs and OHCs were counted in these digital images using a scale of 100 µm and the Cell_Count plugin.

The hair cells were classified into three morphological groups: intact, damaged, and missing. Intact hair cells were characterized by physiologically arranged stereocilia (Figure 4b). Damaged hair cells were characterized by disrupted stereocilia arrangement and/or fusion of the stereocilia (Figure 4c). Missing hair cells were identified by the lack of stereocilia and possible scarring. Occasionally, either the fourth row of OHCs was observed or a preparation caused damage to the hair cells. Such areas were excluded from the count. The mean value of intact, defective, and missing IHCs and OHCs was then reported in 10% intervals over the entire length of the OC.

#### 4.4.2. Spiral Ganglion Neuron Counting

Two methods were used to quantify the SGN. The first method was direct quantification, whereas the second method was semi-quantitative. For both scenarios, the SGNs were labeled using antibodies against neurofilament 200 and a secondary antibody that was Alexa-Fluor-488-conjugated. The images were captured under 40× magnification. Since the SGNs are present in various planes of the cochlear tissue, several images from different planes were merged into a Z stack in ImageJ. The SGN numbers were determined using the first method in digital images on a scale of 100 µm. The Cell_Count plugin was used for cell counting. For the semi-quantitative evaluation, an area of 100 µM length was selected between the habenula perforata and IHCs (ROI) and used to determine the area fraction of the SGN. The evaluation was performed by dividing the cochlea into 10 sections. For each section, two areas were selected as representative and defined as ROI. The fluorescent signal was converted into a binary image and the area fraction of the SGN in the ROI was given as a percentage.

#### 4.4.3. Mast Cell Counting

For the distributive evaluation of the MCs, the OC was divided into hair cell area, neurite area, and SG cell body area. The lateral walls were divided into SV and SL. The granulation status of the MCs was based on Ribatti’s (2018) classification [106]. A distinction was made between non-degranulated MCs and degranulated MCs. Non-degranulated MCs were characterized by the fact that no granules could be found in the extracellular space, and the MCs thus appeared compact and granular. Degranulated MCs were surrounded by numerous extracellular granules and/or had a reduced intracellular granule content. The MCs were counted in the entire OC and LW in 400-fold enlarged images and then summarized.

### 4.5. Statistical analyses

The statistical analyses were performed using the software GraphPad Prism 8 (Version 8.4.3, GraphPad Software, Inc., San Diego, CA, USA (accessed on 4 November 2021)). The two-way ANOVA analysis of variance with multiple post hoc comparisons using the Dunnett test calculated the statistical significance. If the conditions for normal distribution were not met, a Kruskal–Wallis test with multiple post hoc comparisons was carried out using Dunn’s test. A significance value (*p*-value) below 0.05 was considered significant and *p* > 0.05 considered insignificant. The results of the descriptive statistics were presented as the mean ± standard error of the mean (SEM).

## 5. Conclusions

Our results demonstrate that murine cochlear mast cells degranulate after 24 h of exposure to cisplatin and that cromolyn can prevent that process. Furthermore, cromolyn used at 5 or 25 µM significantly inhibited cisplatin-induced auditory hair cell damage, whereas cromolyn used at 10 µM inhibited spiral ganglion loss caused by cisplatin in the cochlea. Further research on using cromolyn or other mast cell stabilizers to protect from cisplatin ototoxicity should offer possible new therapeutic approaches for attenuating hearing loss induced by that chemotherapeutic approach.

## Figures and Tables

**Figure 1 ijms-24-04620-f001:**
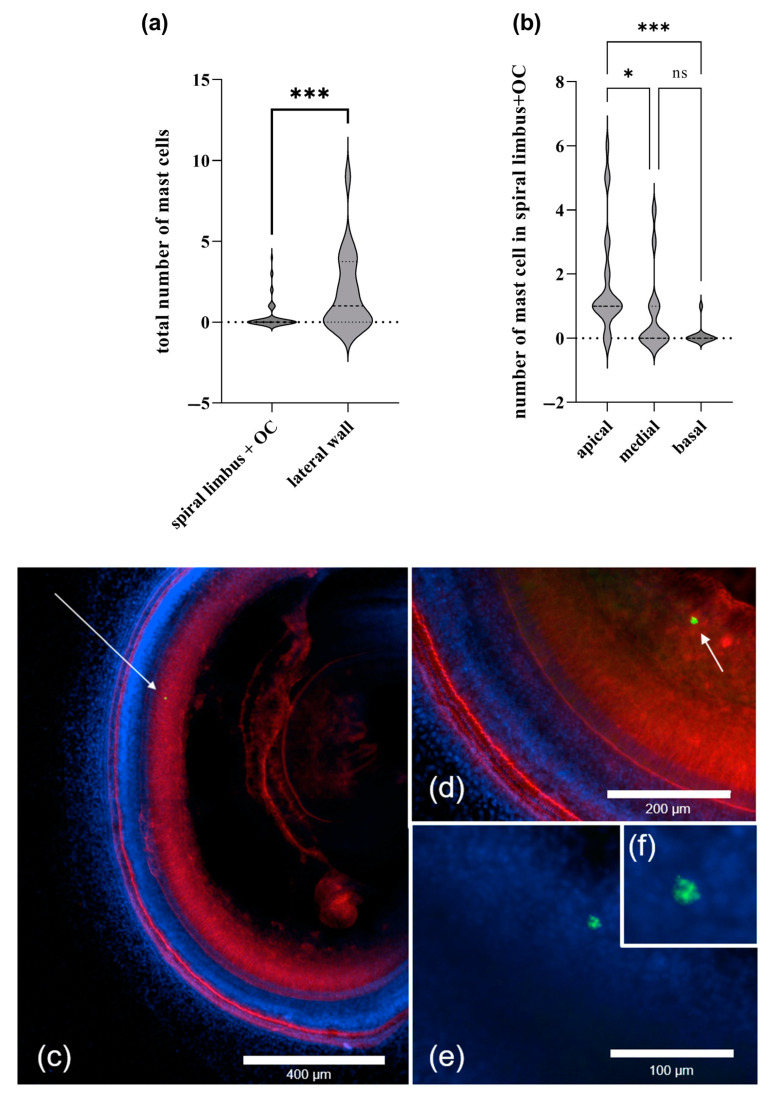
Distribution of MCs in the membranous cochlea. The MCs were scored in each cochlear preparation, and the results are presented as a total number of MCs (non-degranulated and degranulated) per area ((**a**) *n* = 7; (**b**) *n* = 9). (**a**) Violin plot demonstrating MCs distribution in the explant areas containing either spiral limbus with OC or the lateral wall; (**b**) Violin plot demonstrating MCs distribution in the apical, medial, and basal parts of the cochlea; the marker shows the mean. (**c**–**f**) Representative micrographs showing the cochlear MCs in the medial part of the cochlea. The explants were stained with phalloidin-iFluor 594, avidin–Alexa Fluor ™ 488, and DAPI. The arrows point to MCs. (**f**) contains a digital enlargement of the MC image form (**e**). The data were derived from two independent experiments, and the differences were calculated for means. “ns” indicates not significant (*p* > 0.05); * *p* < 0.05; *** *p* < 0.001. ((**a**) Kruskal–Wallis test; (**b**) Mann–Whitney test).

**Figure 2 ijms-24-04620-f002:**
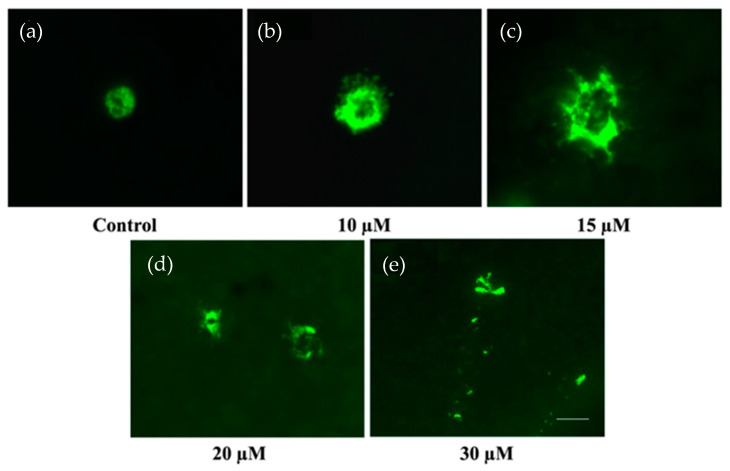
Representative epifluorescence micrographs showing the cochlear MCs in the LW after 24 h of exposure to 10 µM, 15 µM, 20 µM, and 30 µM cisplatin. The MCs were labeled with avidin–Alexa Fluor ™ 488. (**a**) Cochlear MC after incubation in the culture medium only (control). (**b**) Cochlear MC after exposure to 10 µM cisplatin. (**c**) Cochlear MC after exposure to 15 µM cisplatin. (**d**) Cochlear MC after exposure to 20 µM cisplatin. (**e**) Cochlear MC after exposure to 30 μM cisplatin. The scale bar represents 10 µm.

**Figure 3 ijms-24-04620-f003:**
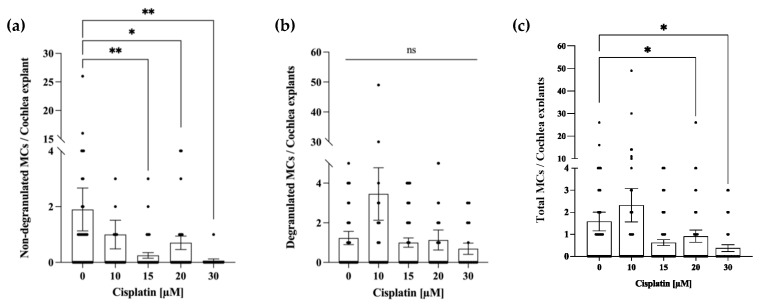
Cisplatin decreases the number of non-degranulated MCs and causes a reduction of cochlear MCs at higher concentrations. The control group (n = 9) was cultured for 24 h in a culture medium. The remaining groups were exposed to 10 µM (n = 10), 15 µM (n = 4), 20 µM (n = 12), or 30 µM (n = 4) of cisplatin for 24 h. (**a**) The absolute number of non-degranulated MCs per explant (spiral limbus and LW). (**b**) The absolute number of degranulated MCs per explant (spiral limbus and LW). (**c**) The total number of MCs per explant (containing spiral limbus with OC and LW as well non-degranulated and degranulated MCs). The average number of MCs per naive explant was 1.58 ± 0.42. The data represent four independent experiments and are reported as a mean ± SEM. “ns” indicates not significant (*p* > 0.05); * *p* < 0.05; ** *p* < 0.01; (Kruskal–Wallis test with Dunn’s multiple comparison test).

**Figure 4 ijms-24-04620-f004:**
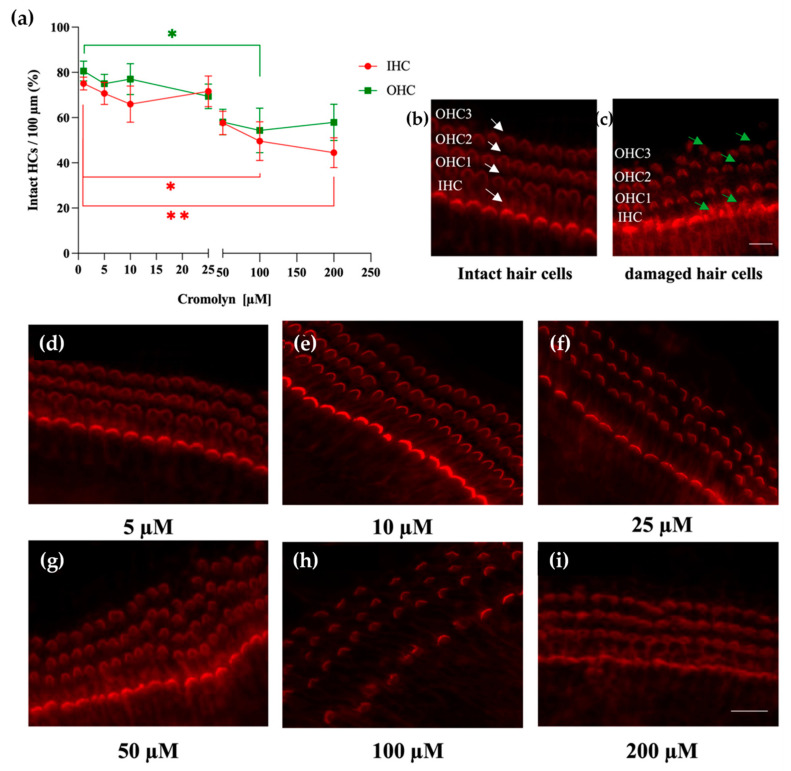
Cromolyn used at high concentrations decreases the numbers of IHCs and OHCs. (**a**) The intact hair cells were scored along the length of 100 µm of the cochlear explant (spiral limbus containing OC), and the percentages of intact IHCs (red circles) and OHCs (green squares) were determined and plotted on the *y*-axis. The control explants (n = 4) were cultured for 24 h in a tissue culture medium. The treatment groups (n = 4 for each treatment) were cultured for 24 h with cromolyn at the following concentrations: 5 µM, 10 µM, 25 µM, 50 µM, 100 µM, and 200 µM. (**b**,**c**) Representative micrograph showing intact (**b**) and damaged (**c**) hair cells. Arrows point out intact HCs (white) and damaged HCs (green). Scale bar represents 10 µM. (**d**–**i**) Representative micrograph showing the HCs after 24 h of exposure to 5 µM, 10 µM, 25 µM, 50 µM, 100 µM, and 200 µM cromolyn. The cochlea explants were stained with phalloidin-iFluor 594. The scale bar represents 10 µm. Four independent experiments were performed; the data are reported as mean ± SEM. * *p* < 0.05; ** *p* < 0.01 (two-way ANOVA with Dunnett multiple comparison test).

**Figure 5 ijms-24-04620-f005:**
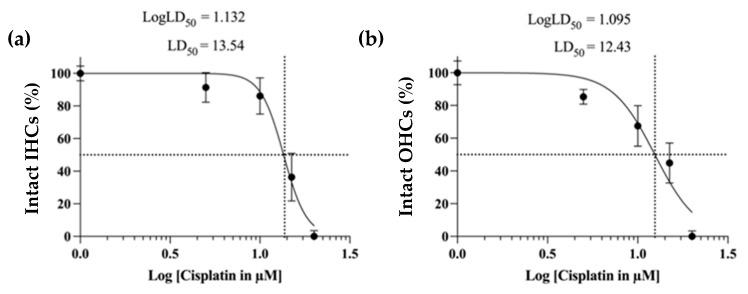
Dose effect of cisplatin as a function of the percentage of intact IHCs and OHCs. The cell survival rate was determined based on the percentage of intact IHCs and OHCs. For this purpose, the intact HCs (n = 4 each) were quantified lengthwise in the entire OC (per 100 µm) in specimens exposed to 5 µM, 10 µM, 15 µM, and 20 µM cisplatin. The control group (n = 3) was cultured in a standard medium for 24 h. Shown is the percentage of intact HCs in the entire OC. The values were normalized, and the cisplatin concentrations (abscissa) were presented on a logarithmic scale. (**a**) Dose–response curve for the intact IHCs. (**b**) Dose–response curve for the intact OHCs. The data represent four independent experiments and are reported as mean ± SEM.

**Figure 6 ijms-24-04620-f006:**
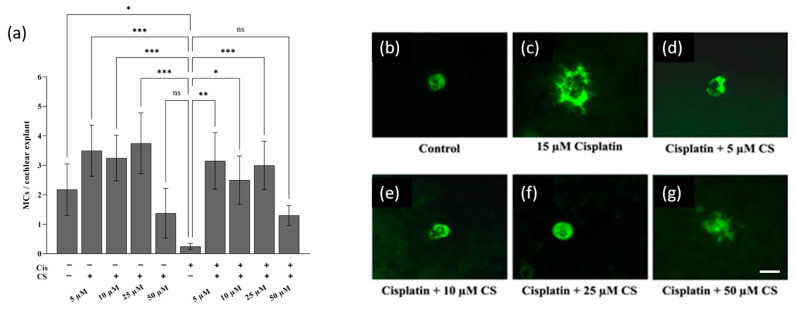
Cromolyn prevents cisplatin-induced MC degranulation in cochlear explants. (**a**) The control groups (n = 4 each) were either not treated (− −) or incubated for 24 h with 5, 10, 25, or 50 µM of cromolyn (CS, − +) and exposed to 15 µM cisplatin (Cis, + −). The treatment groups (n = 4 each) were pretreated with 5, 10, 25, and 50 µM CS for 2 h and then exposed to 15 μM cisplatin for 24 h (+ +). The non-degranulated MCs were counted in the explant tissues. (**b**–**g**) Representative micrographs show MCs in the respective treatment groups. The MCs were labeled with avidin–Alexa Fluor ™ 488. The scale bar represents 10 µm. The data were derived from four independent experiments and are reported as mean ± SEM. “ns” indicates not significant (*p* > 0.05); * *p* < 0.05; ** *p* < 0.01; *** *p* < 0.001 (Kruskal–Wallis test with Dunn’s multiple comparison test).

**Figure 7 ijms-24-04620-f007:**
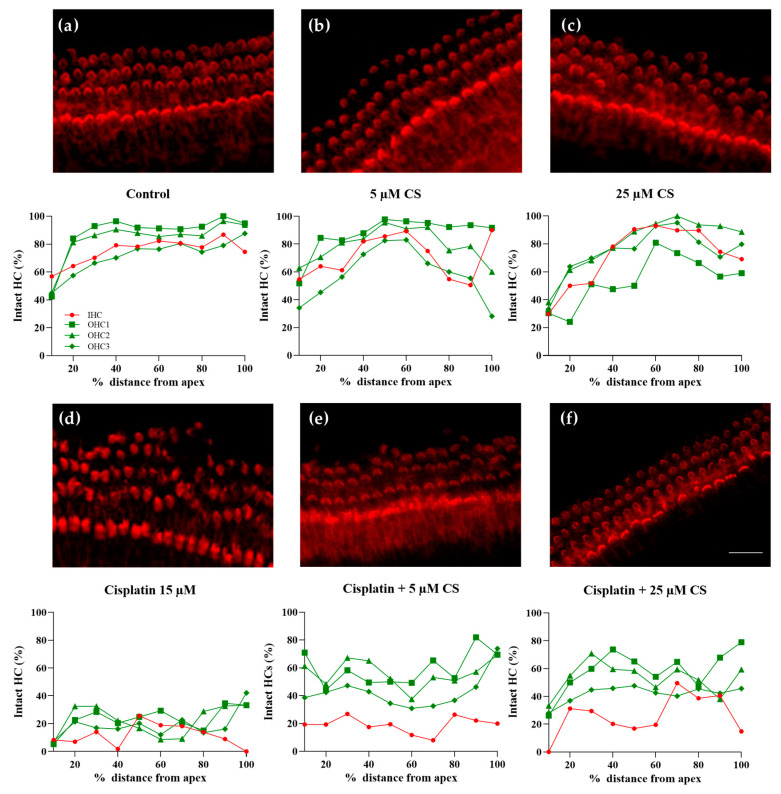
Cromolyn protects the cochlear tissues from cisplatin-induced hair cell damage and loss. The representative epifluorescence images of phalloidin-iFluor-594-stained cochlear explants and the representative cochleograms from each experimental condition are presented. In the cochleograms, the mean percentage per section of intact hair cells is plotted in relation to the distance from the base to the apex. The cochlea was divided into 10 sections for scoring, each representing 10% of the total OC length (0% = apex − 100% = base). (**a**) A typical hair cell array was observed in the control explants. (**b**) Cromolyn (CS) used at 5 µM does not affect the HCs. (**c**) Cromolyn at 25 µM does not affect the HCs. (**d**) Exposure to 15 µM of cisplatin (Cis) reduces the percentage of intact HCs. (**e**) Cromolyn used at 5 µM or (**f**) 25 µM protects the HCs from the cisplatin-induced damage. The scale bar represents 10 µm. For group statistics, see Figure 8.

**Figure 8 ijms-24-04620-f008:**
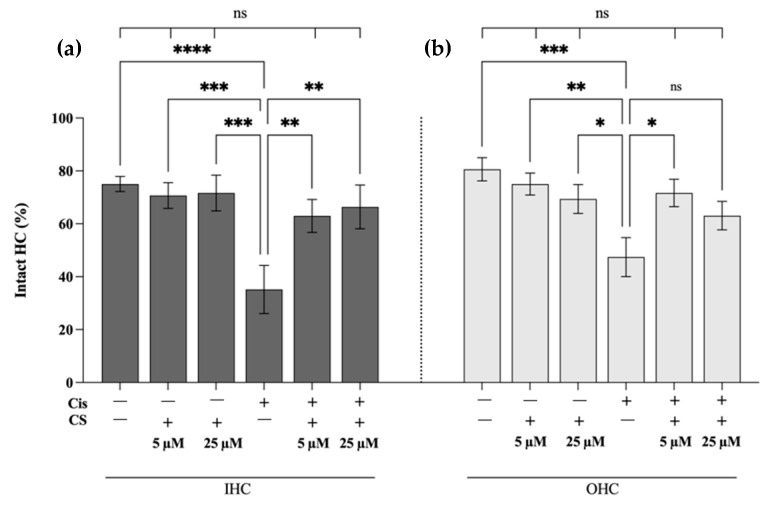
Cromolyn prevents the cisplatin-induced loss of auditory hair cells. The control groups (n = 4 each) were either not treated (− −) or incubated for 24 h with 5 µM or 25 µM of cromolyn (CS, − +). The treatment groups (n = 4 each) were pretreated with 5 μM and 25 μM CS for 2 h and then exposed to 15 μM cisplatin for 24 h (+ +). Both the control and treatment groups were compared to the cisplatin group (+ −). (**a**) The percentage of intact IHCs in control and treatment groups. (**b**) The percentage of intact OHCs in control and treatment groups. The data were derived from four independent experiments and are reported as mean ± SEM. “ns” indicates not significant (*p* > 0.05); * *p* < 0.05; ** *p* < 0.01; *** *p* < 0.001; **** *p* < 0.0001 (two-way ANOVA with Dunnett multiple comparison test).

**Figure 9 ijms-24-04620-f009:**
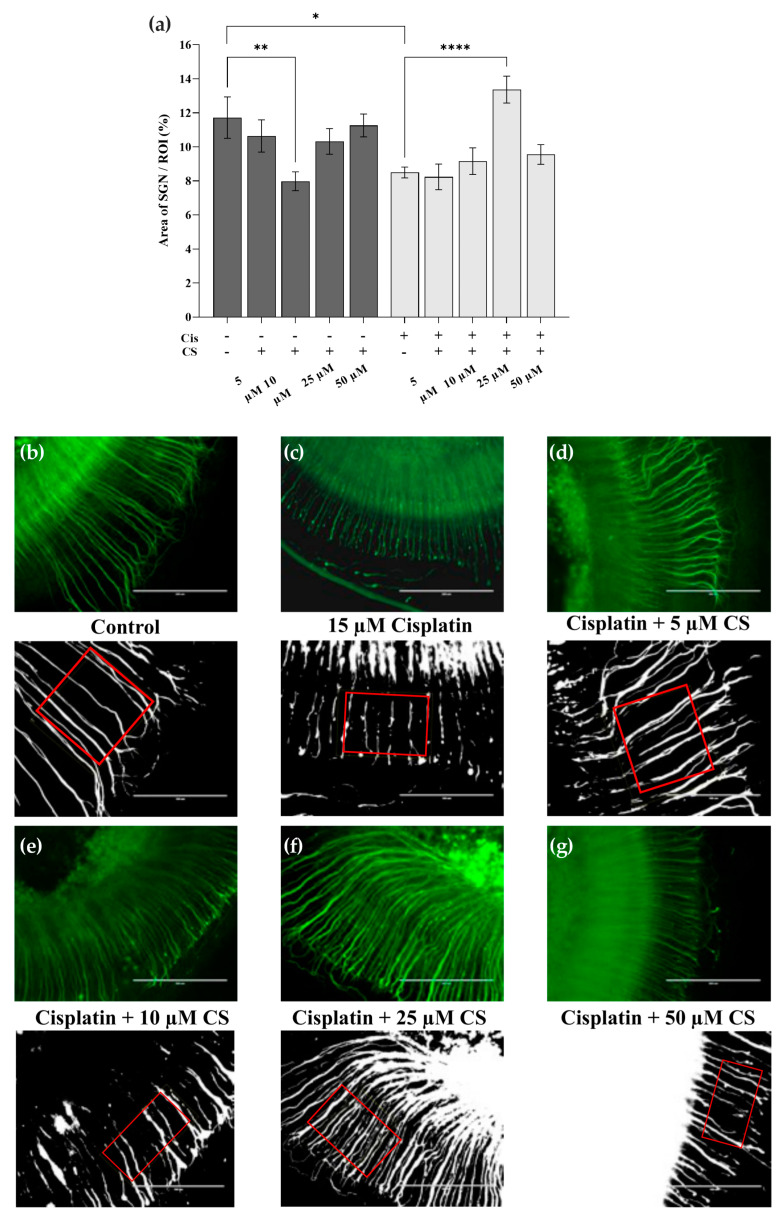
Pretreatment of cochlear explants with 25 µM cromolyn protects SGN from cispla tin-induced neurotoxicity. The control groups (n = 4 in each case) were either untreated or incubated with CS (5, 10, 25, 50 μM) for 24 h. The treatment groups (n = 4 in each case) were pretreated with 5, 10, 25, and 50 μM CS for 2 h and then exposed to 15 μM cisplatin for 24 h. The control and treatment groups were compared to the cisplatin group. The explants were labeled with NF200 and a secondary antibody conjugated with the Alexa FluorTM 488. The areas containing the SGN were quantified in five representative sections of the apical, medial, and basal fragments. The ROI area was measured between the habenula perforata and the IHC and given as a percentage. (**a**) Between-group differences in ROI. (**b**–**g**) Representative micrographs demonstrate SGNs in the respective treatment groups and the associated binary images. The ROI areas are outlined in red. The scale bar in the SGN images represents 200 µm and 100 µm in the binary images. The data represent four independent experiments and are reported as mean ± SEM. * *p* < 0.05; ** *p* < 0.01; **** *p* < 0.0001 (two-way ANOVA with Dunnett multiple comparison test).

**Figure 10 ijms-24-04620-f010:**
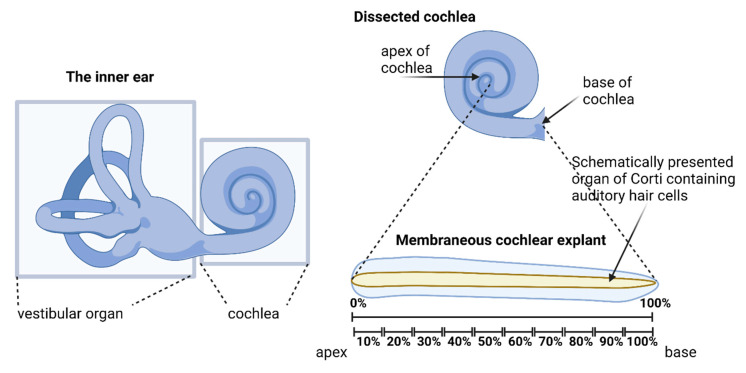
Illustration of the hair cell scoring approach along the cochlear length. That approach was used to deliver the cochleogram data presented in Figure 7. Created with BioRender.com.

## Data Availability

Data are available on request.

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
