# Peer review of "Degranulation of Murine Resident Cochlear Mast Cells: A Possible Factor Contributing to Cisplatin-Induced Ototoxicity and Neurotoxicity"

_ijms, 2023, doi:10.3390/ijms24054620_

Round 1

Reviewer 1 Report

In this paper, the author wants to follow on the recently identified resident mast cells in rodents’ cochleae and observed that the number of mast cells changed upon adding cisplatin to cochlear  explants as  they followed the observation and found that the murine cochlear mast cells degranulate in response to cisplatin and that the mast cell stabilizer cromoglicic acid (cromolyn) inhibits this process. Additionally, cromolyn significantly prevented cisplatin-induced loss of auditory hair cells and spiral ganglion neurons. This manuscript is generally well written. However Strengths and limitations of our study? What should be focus on in future research?

Author Response

We thank the Reviewer for the time and thought spent on our manuscript. Reviewer's remarks are appreciated. Below are the point-to–point replies to the comments.

However Strengths and limitations of our study?

Thank you for the comment. In response to it, we changed the former subsection title "3.6. Study limitations" to "3.6. Study strengths and limitations" and mention now the strengths:

"This system allows studying the intact morphology of the auditory periphery, which is a strength of our study."

"Still, working with native cochlear MC is seen as a strength of this study."

What should be focus on in future research?

In response to this comment, we added the following subsection:

"3.7. Future directions

The present work demonstrated that cromolyn could significantly prevent the cisplatin-induced MCs degranulation and the oto- and neurotoxicity in the murine cochlear explants. This finding implies cochlear mast cells' possible involvement in cisplatin-induced cochlear toxicity. Further in vitro and in vivo preclinical experiments need to verify obtained results. Future experiments using mast cells purified from the peritoneum or bone marrow of experimental animals would be of special importance. Such an approach would enable precise measurement of the released mediators and allow for performing the time course experiments. Another extension of the present work would be a clinical trial in which patients receive cromoglicic acid before cisplatin treatment, and their hearing is subsequently assessed during therapy.

Furthermore, the actual mechanism in which MC contribute to cisplatin-induced ototoxicity needs to be explored by investigating the released mediators and the bio-chemical and molecular processes induced in the cochlea by MC degranulation. That could result in promising therapeutic approaches for treating cisplatin-induced hear-ing loss. Additionally, an extended collaboration between otorhinolaryngologists, immunologists, and dermatologists is advocated to address further questions and improve patient welfare.

Lastly, particular attention should be paid in the future to the prevention of mitochondrial dysregulation, which has not been studied here. Such dysregulation has been described as the primary mechanism in the pathogenesis of cisplatin-induced toxicity [98]. In addition, cisplatin can accumulate in the mitochondrial matrix and negatively affect the antioxidant system, decreasing intracellular ATP and inducing ROS production, which causes mitochondrial dysfunction and activates apoptotic pathways [24,39-41]. Interestingly, it has been shown that the levels of peroxisome proliferator-activated receptor gamma coactivator 1-alpha (PGC-1α), a protein that regulates mitochondrial biogenesis [99], decrease during this process [100]. Polyphenols, the largest group of phytochemicals, were demonstrated to regulate mitochondrial biogenesis, positively influence mitochondrial processes, and have anti-neurodegenerative properties [101,102]. Therefore, it is tempting to suggest using poly-phenols (e.g., curcumin or resveratrol) during chemotherapy with cisplatin to protect the auditory periphery from the sensorineurodegenerative effects of cisplatin, particularly because these agents can synergize in the potentiating anti-tumor effect of other medications [103]. Given that polyphenols can decrease mast cell activation and degranulation [104], they may offer in the future natural dietary protection from various pathogenic mechanisms induced by cisplatin in the inner ear."

Reviewer 2 Report

1.It is suggested to add more in vivo and clinical studies in introduction part.

2.what is the suggestion of this study for future works?

3.Please discuss and compare your results with previous works and add suggestions.

4.It will be better to add the role of mitochondria.

5.Please add role of cell junctions proteins.

6.More references for the discussion part of manuscript and bold your study novelty should be added: e.g.,

-DOI: 10.1016/j.chemosphere.2022.136020

-DOI: 10.1155/2021/4946711

Author Response

We thank the Reviewer for the time and thought spent on our manuscript. Following the Reviewers’s recommendations, we extended the Introduction and added there the available clinical and basic research that strengthens the justification of our research. We also added the requested references and extended the reference list in general. Lastly, we revised the Methods. Below you will find poin-to-point response to the detailed suggestions.

1.It is suggested to add more in vivo and clinical studies in introduction part.

We have included the following passages in the Introduction:

"Finally, functional studies using the guinea pig model of type-I allergy demonstrated allergen-induced MC degranulation in the endolymphatic sac concurrent with nystagmus [5,6] and temporary hearing loss. These symptoms could be prevented by an inhibitor of MC histamine release, tranilast [7], or MC stabilizer pemirolast [8]."

" Clinical evidence links mast cell disorders to inner ear illnesses. In patients with con-genital or familial mastocytosis, sensorineural (cochlear) hearing loss was diagnosed [11,12]. Furthermore, sensorineural hearing loss is common among patients with allergic rhinitis, and MC involvement was suggested as a possible mechanism [12]. Finally, in a patient with cutaneous mastocytosis and Meniere disease, therapy with omalizumab relieved the mastocytosis symptoms and those caused by Meniere disease affecting the inner ear [13].”

2.what is the suggestion of this study for future works?

To address this comment, we added the following subsection in the Discussion:

"3.7. Future directions

The present work demonstrated that cromolyn could significantly prevent the cisplatin-induced MCs degranulation and the oto- and neurotoxicity in the murine cochlear explants. This finding implies cochlear MCs possible involvement in cispla-tin-induced cochlear toxicity. Further in vitro and in vivo preclinical experiments need to verify obtained results. Future experiments using MCs purified from the peritone-um or bone marrow of experimental animals would be of special importance. Such an approach would enable precise measurement of the released mediators and allow for performing the time course experiments. Another extension of the present work would be a clinical trial in which patients receive cromoglicic acid before cisplatin treatment, and their hearing is subsequently assessed during therapy.

Furthermore, the actual mechanism in which MC contribute to cisplatin-induced ototoxicity needs to be explored by investigating the released mediators and the biochemical and molecular processes induced in the cochlea by MC degranulation. That could result in promising therapeutic approaches for treating cisplatin-induced hearing loss. Additionally, an extended collaboration between otorhinolaryngologists, immunologists, and dermatologists is advocated to address further questions and improve patient welfare.

Lastly, particular attention should be paid in the future to the prevention of mitochondrial dysregulation, which has not been studied here. Such dysregulation has been described as the primary mechanism in the pathogenesis of cisplatin-induced toxicity [98]. In addition, cisplatin can accumulate in the mitochondrial matrix and negatively affect the antioxidant system, decreasing intracellular ATP and inducing ROS production, which causes mitochondrial dysfunction and activates apoptotic pathways [24,39-41]. Interestingly, it has been shown that the levels of peroxisome proliferator-activated receptor gamma coactivator 1-alpha (PGC-1α), a protein that regulates mitochondrial biogenesis [99], decrease during this process [100]. Polyphenols, the largest group of phytochemicals, were demonstrated to regulate mitochondrial biogenesis, positively influence mitochondrial processes, and have anti-neurodegenerative properties [101,102]. Therefore, it is tempting to suggest using poly-phenols (e.g., curcumin or resveratrol) during chemotherapy with cisplatin to protect the auditory periphery from the sensorineurodegenerative effects of cisplatin, particularly because these agents can synergize in the potentiating anti-tumor effect of other medications [103]. Since polyphenols can decrease MC activation and degranulation [104], they may offer natural dietary protection from various pathogenic mechanisms induced by cisplatin in the inner ear."

3.Please discuss and compare your results with previous works and add suggestions.

We discussed and compared the results with the result of others in the Discussion:

"Our earlier work has shown that exposure of cochlear explants to 20 µM cisplatin paradoxically increases the total number of cochlear MCs in Wistar rats (9). In the present work, upon exposure to 20 µM cisplatin, a significant decrease in the number of MCs could be observed. One possible reason for this discrepancy is using different animal species and a distinct postnatal developmental stage – in the previous work, 5-day-old Wistar rats were used, while here, we used 3-5 days-old C57BL/6 mice. The observation of increased MC numbers upon exposure to cisplatin is supported by Takagi et al., who also observed an increased number of MCs after activation with IgE and attributed it to the presence of undifferentiated MCs, which began to differentiate after stimulation (47). The proliferation and differentiation of the MCs occur in the presence of stem cell factor (SCF), which can be produced by resident cochlear macrophages (48) and, similarly to TNF-alpha and IL-1 beta (49), released upon exposure to cisplatin. Future studies should expand this observation."

4.It will be better to add the role of mitochondria.

We added the following paragraph in the Discussion section addressing the role of mitochondria in cisplatin toxicity and possible strategies (that include using polyphenols) to prevent these detrimental events from occurring:

"Lastly, particular attention should be paid in the future to the prevention of mitochondrial dysregulation, which has not been studied here. Such dysregulation has been described as the primary mechanism in the pathogenesis of cisplatin-induced toxicity [94]. In addition, cisplatin can accumulate in the mitochondrial matrix and negatively affect the antioxidant system, decreasing intracellular ATP and inducing ROS production, which causes mitochondrial dysfunction and activates apoptotic pathways [24,39-41]. Interestingly, it has been shown that the levels of peroxisome proliferator-activated receptor gamma coactivator 1-alpha (PGC-1α), a protein that regulates mitochondrial biogenesis [95], decrease during this process [96]. Polyphenols, the largest group of phytochemicals, were demonstrated to regulate mitochondrial biogenesis, positively influence mitochondrial processes, and have anti-neurodegenerative properties [97,98]. Therefore, it is tempting to suggest using polyphenols (e.g., curcumin or resveratrol) during chemotherapy with cisplatin to protect the auditory periphery from the sensorineurodegenerative effects of cisplatin, particularly because these agents can synergize in the potentiating anti-tumor effect of other medications [99]. Given that polyphenols can decrease mast cell activation and degranulation [100], they may offer in the future natural dietary protection from various pathogenic mechanisms induced by cisplatin in the inner ear. "

5.Please add role of cell junctions proteins.

We followed this suggestion and added the following to the Discussion:

"The last possible mechanism in which cisplatin-mediated degranulation of MC could cause cochlear damage is proteolytic damage to the cochlear tissues and interference. Mast cell granules contain several proteases [54] that can target tight junctions disrupting epithelial or connective tissue barriers [55,56]. Since connexins and claudins representing gap junction proteins are indispensable in the inner ear for potassium recycling and maintaining endocochlear potential, loss or damage of junction proteins could result in hearing loss, as demonstrated by hereditary human deafness associated with mutations affecting connexin 26, 29, 30, 31, and 43 [57]. However, in the present work, the issue of tight junctions and MC proteases was not studied because of the model we used. It should, however, be addressed in the future using an animal model."

6.More references for the discussion part of manuscript and bold your study novelty should be added: e.g.,

-DOI: 10.1016/j.chemosphere.2022.136020

-DOI: 10.1155/2021/4946711

We included and discussed the recommended literature ( DOI: 10.1016/j.chemosphere.2022.136020; DOI: 10.1155/2021/4946711) and expanded the existing in the Discussion and Introduction.

Reviewer 3 Report

The authors analyzed degranulation of murine cochlear mast cells and the effect of cromolyn on them. The paper is somewhat interesting. However, I have both major and minor comments.

Major comments.

Line 111.

Did the authors assess mast cell degranulation following shorter (approximately 1 hour) incubation with cisplatin? Immediate degranulation may be an important and typical feature of mast cells.

Figure 9a.

As the authors mention in the Discussion section, the results that only one concentration of CS demonstrated significant effects are strange. Were the results reproducible? How many separate experiments did the authors try? When the authors added cisplatin solution following 2-hour pretreatment with CS, was CS diluted to lower concentrations?

Lines 590 - 591.

The phrase “cromolyn significantly inhibits the ototoxicity caused by cisplatin in the cochlea” needs to be changed, since the authors analyzed the effect of cromolyn on each cell biology, but not on the whole ototoxicity due to cisplatin.

Minor comments.

Lines 330 - 331.

TNF-a should be changed to TNF-alpha. IL-1b should be changed to IL-1beta.

Author Response

Reviewer's comments and suggestions are appreciated. Prompted by the remarks, we modified the conclusions of our study:

“Our results demonstrate that murine cochlear mast cells degranulate after 24 hours of exposure to cisplatin and that cromolyn can prevent that process. Furthermore, cromolyn used at 5 or 25 µM significantly inhibited cisplatin-induced auditory hair cell damage, whereas cromolyn used at 10 µM inhibited spiral ganglion loss caused by cisplatin in the cochlea. Further research on using cromolyn or other mast cell stabilizers to protect from cisplatin ototoxicity should offer possible new therapeutic approaches for attenuating hearing loss induced by that chemotherapeutic.”

Other comments are addressed point-to-point below.

The authors analyzed degranulation of murine cochlear mast cells and the effect of cromolyn on them. The paper is somewhat interesting. However, I have both major and minor comments.

Major comments.

Line 111.

Did the authors assess mast cell degranulation following shorter (approximately 1 hour) incubation with cisplatin? Immediate degranulation may be an important and typical feature of mast cells.

We did not assess the degranulation at early time points. However, upon reading the comment, we conducted an additional experiment to address this point and exposed the cochlear explants to 15 or 30 µM cisplatin for 90 min. However, the results were ambiguous, so we have not included them in the manuscript or supplementary information.

For the information of the Reviewer: in the two control cochleae, we detected two mast cells per each explant; one was degranulated and the other not. In the explants treated with 15 µM cisplatin: one contained four mast cells (3 intact and one degranulated), whereas the other was mast cell-free. In the explants treated with 30 µM cisplatin: one contained two mast cells (both were degranulated) and the other one – one mast cell (also degranulated). Unfortunately, we had no more animals this week and could not perform more experiments, which would be necessary to reach statistically meaningful conclusions.

We plan to address this question in the future using BMMC cultures by measuring the concentration of mast cell mediators in the supernatant (e.g., β-hexosaminidase, chymase, or tryptase) of cisplatin-exposed cells. In addition to cisplatin, which we will titrate, we will also use compound 48/80 and cross-linking of IgE as controls.

Figure 9a.

As the authors mention in the Discussion section, the results that only one concentration of CS demonstrated significant effects are strange.

The comment is appreciated. Concerning the hair-cell loss induced by cisplatin, CS was effective when used at two concentrations (5 and 25 µM), as mentioned in paragraph 3.4. “Effect of cromolyn on cisplatin-induced hair cell damage”. Regarding the SGN loss induced by cisplatin, we showed that cromolyn is effective only at one concentration of 25 µM. We supported our observations with the following lines of thinking: "During the experiments where various cromolyn concentrations were used to determine possible effects on the SGNs, we observed a significant drop in the neurite density when using 10 µM but not lower (5 µM) or higher (25, or 50 µM) cromolyn concentrations, suggesting a nonlinear response to cromolyn. Cromolyn binds to proteins from the S100 family [63], expressed in the developing cochlear tissues in mice on the hair cells, supporting cells, and spiral limbus [64]. Since the Ca2+-binding S100 can regulate a wealth of physiological intracellular and extracellular processes, including neurotrophy [65], it is plausible that cromolyn at a particular concentration might interfere with that process in the still-developing cochlea. Another target of cromolyn is the G-protein coupled receptor 35 (GPR35). Recent studies suggested that cromoglicic acid acts as a selective agonist of GPR35, modulating the intracellular Ca2+ release [66,67]. The expression pattern of GPR35 in the cochlea is unknown, but it is known that various immune cells, including mast cells, express the GPR35 on their surface [66,68]. Here, too, the nonlinear type of response to cromolyn is puzzling and requires further investigation."

Were the results reproducible? How many separate experiments did the authors try?

The results were reproducible. We performed four separate experiments for each concentration to address the experimental reproducibility.

When the authors added cisplatin solution following 2-hour pretreatment with CS, was CS diluted to lower concentrations?

After two hours of pretreatment with CS and before adding cisplatin, the explants were washed two times with tissue culture medium, 5 minutes each wash. Thus, we assume that no residual CS was left in the medium.

We revised the Methods and added the following information: "In the experiments with cisplatin exposure, the explants were preincubated with 5 μM, 10 μM, 25 μM, or 50 μM cromolyn for 2h, then rinsed twice with DMEM / F12 for 5 minutes and then exposed to 15 μM cisplatin for 24h."

Lines 590 - 591.

The phrase "cromolyn significantly inhibits the ototoxicity caused by cisplatin in the cochlea" needs to be changed, since the authors analyzed the effect of cromolyn on each cell biology, but not on the whole ototoxicity due to cisplatin.

The comment is appreciated. We changed the phrase to:

"Furthermore, cromolyn significantly inhibits the hair-cell and spiral ganglion loss caused by cisplatin in the cochlea."

Minor comments.

Lines 330 - 331.

TNF-a should be changed to TNF-alpha. IL-1b should be changed to IL-1beta.

Changed.

Reviewer 4 Report

This study investigates how murine cochlear mast cell degranulation induced by cisplatin may contribute to its oxotoxic effects, and how treatment with cromolyn may prevent this. This is an interesting study, and may lead to potential treatments/preventions for chemotherapy-induced hearing loss, however some further explanations/experiments may be needed to fully confirm the conclusions the authors have made in the study. 

Line 90: Figures 1a and 1b have been switched around in the text, please fix

Figure 3: In this figure, the authors show a signficant decrease in the numbers of non-degranulated mast cells but no significant increase in the numbers of degranulated mast cells when treated with cisplatin. The authors have not included a total number of mast cells to see if this decrease in the non-degranulated mast cells is due to a loss/death of the mast cells rather than degranulation. Also, perhaps this is not the best stain to demonstrate mast cell degranulation - why did the authors choose avidin stain over the more widely used toludine blue? 

Figure 6: The authors here use the numbers of non-degranulated mast cells again as a measure of degranulation - however in Figure 3 as stated above, there was no corresponding increase in degranulated mast cells with the decrease in non-degranulated mast cells, so how can the authors be sure that these mast cells have degranulated rather than died? No total numbers of mast cells have been given here either, just non-degranulated numbers, so it is difficult to say whether the decrease in non-degranulated mast cells is due to degranulation or death from a readers perspective. 

It would be nice to see a study of cisplatin and cromolyn on mast cells in vitro, however I understand this can be difficult and may not be possible to include in this study if the authors do not have previous experience in culturing murine or human mast cells. 

Figure 8: How were the statistics/numbers calculated from the graphs in Figure 7? Were all the numbers of intact HCs added up at each distance from the apex? It is not clear. 

Figure 9: It is unusual that only very specific concentrations of CS have an effect - i.e. 10uM but not 5uM or 25uM decreased SGN and only 25uM restored SGN after cisplatin treatment. How were the ROIs chosen? Were multiple ROIs in the same image chosen and an average number calculated or one single ROI? 

Author Response

We appreciate the constructive criticism and the Reviewer's comments and suggestions. We revised the manuscript paying particular attention to better explaining the research design, and modified the Conclusions:

“Our results demonstrate that murine cochlear mast cells degranulate after 24 hours of exposure to cisplatin and that cromolyn can prevent that process. Furthermore, cromolyn used at 5 or 25 µM significantly inhibited cisplatin-induced auditory hair cell damage, whereas cromolyn used at 10 µM inhibited spiral ganglion loss caused by cisplatin in the cochlea. Further research on using cromolyn or other mast cell stabilizers to protect from cisplatin ototoxicity should offer possible new therapeutic approaches for attenuating hearing loss induced by that chemotherapeutic.”

Below you will find a point-to-point answer to the particular points made:

Line 90: Figures 1a and 1b have been switched around in the text, please fix

We have corrected that.

Figure 3: In this figure, the authors show a signficant decrease in the numbers of non-degranulated mast cells but no significant increase in the numbers of degranulated mast cells when treated with cisplatin. The authors have not included a total number of mast cells to see if this decrease in the non-degranulated mast cells is due to a loss/death of the mast cells rather than degranulation.

We added the respective graph showing the total number of MCs In Figure 3c, with the following paragraph in the Results.

"Also, as demonstrated in Figure 3c, the number of MCs in the explants decreased with increasing cisplatin concentration (OC and LW combined). On average, there were 1.58 ± 0.42 MCs in an untreated explant, regardless of the MCs degranulation stage. After exposure to 15 µM cisplatin, the average number of MCs decreased, but the decrease was significant only in the 20 µM and 30 µM groups. In the group exposed to 10 µM cisplatin, an increase in MCs to 2.31 ± 0.75 was recorded; however, that increase was not statistically significant. "

We have added the following paragraph to the Discussion:

"We also demonstrated that exposure to 20 µM and 30 µM of cisplatin leads to a significant decrease in the total number of MCs, indicating a dose-dependent effect either resulting in degranulation of MCs or inducing apoptosis. In agreement with that, experiments with the human MC line HMC-1 demonstrated that high cisplatin concentrations induce MCs' apoptosis (46). It can be concluded that the effect of cisplatin on MCs is either degranulation or apoptosis, depending on the concentration. Completely degranulated MCs are no longer detected by the staining method with avidin. Although the use of the glycoprotein avidin is an established staining method for identifying MC granules (47), after degranulation, the granules are scattered in the extracellular space, which may result in some MCs remaining undetected and thus neglected during quantification."

Also, perhaps this is not the best stain to demonstrate mast cell degranulation - why did the authors choose avidin stain over the more widely used toludine blue? 

The reason behind using fluorescently-labeled avidin instead of the histochemical method has to do with 3R approach, which is strongly recommended by our local authorities and with the methods already established in the laboratory to study the inner ear biology. Using fluorescent markers, we can label several molecules in the same specimen (e.g., hair cells or spiral ganglion neurons) and gain specific information not only about mast cells but also the neurons and auditory epithelium. In our experimental setup, one explant means sacrificing one animal. Using fluorescent multi-target staining, we substantially reduced the number of animals killed for research. Additionally, avidin staining of mast cells was shown to be equally effective as toluidine blue (Grigorev IP, Korzhevskii DE. Modern Imaging Technologies of Mast Cells for Biology and Medicine (Review). Sovrem Tekhnologii Med. 2021;13(4):93-107. doi: 10.17691/stm2021.13.4.10. Epub 2021 Aug 28.).

We added the following in the body text:

" The glycoprotein avidin conjugated with the fluorescent dye Alexa FluorTM, is frequently used to identify MCs and is an established staining method during fluorescent imaging of mast cells "

Figure 6: The authors here use the numbers of non-degranulated mast cells again as a measure of degranulation - however in Figure 3 as stated above, there was no corresponding increase in degranulated mast cells with the decrease in non-degranulated mast cells, so how can the authors be sure that these mast cells have degranulated rather than died? No total numbers of mast cells have been given here either, just non-degranulated numbers, so it is difficult to say whether the decrease in non-degranulated mast cells is due to degranulation or death from a readers perspective. 

The average total number of mast cells in naïve explants was 1.58 ± 0.42, regardless of granulation/degranulation status. It is now mentioned in the text and the description of the amended Figure 3.

We agree with the Reviewer that we have not addressed the issue of mast cell death in our present work, which is a drawback mentioned now in subsection 3.6. Study strengths and limitations:

“An additional problem, which has not been addressed in the current work, was the possible mast cell death, its type, and the routes leading to it. These types of experiments require greater numbers of mast cells, and other detection methods (for example, flow cytometry) coupled with a time curve and are planned to be performed in the future.”

It would be nice to see a study of cisplatin and cromolyn on mast cells in vitro, however I understand this can be difficult and may not be possible to include in this study if the authors do not have previous experience in culturing murine or human mast cells. 

We appreciate this comment and plan to address this question quantitatively and qualitatively (histamine release, chymase, and tryptase ELISA) using purified mast cells during the next few months. To gain expertise, we started to collaborate with the Institute of Allergology at the Charité.

Figure 8: How were the statistics/numbers calculated from the graphs in Figure 7? Were all the numbers of intact HCs added up at each distance from the apex? It is not clear. 

The graph in Figure 7 visualizes ten cochlear sections of the organ of Corti (0% length = apex - 100% length = base). The inner and outer sensory hair cells were counted, and their mean value was then reported in 10% intervals over the entire length of the Organ of Corti.

Figure 8 shows the average percentage of IHCs and OHCs in the total cochlea explants. For this purpose, the ten sections have been combined, and an average has been calculated.

Figure 9: It is unusual that only very specific concentrations of CS have an effect - i.e. 10uM but not 5uM or 25uM decreased SGN and only 25uM restored SGN after cisplatin treatment.

We agree that this result was unusual, but it was a reproducible finding. We have addressed that in the Discussion:

“During the experiments where various cromolyn concentrations were used to determine possible effects on the SGNs, we observed a significant drop in the neurite density when using 10 µM but not lower (5 µM) or higher (25, or 50 µM) cromolyn concentrations, suggesting a nonlinear response to cromolyn. Cromolyn binds to proteins from the S100 family [71], expressed in the developing cochlear tissues in mice on the hair cells, supporting cells, and spiral limbus [72]. Since the Ca2+-binding S100 can regulate a wealth of physiological intracellular and extracellular processes, including neurotrophy [73], it is plausible that cromolyn at a particular concentration might interfere with that process in the still-developing cochlea. Another target of cromolyn is the G-protein coupled receptor 35 (GPR35). Recent studies suggested that cromoglicic acid acts as a selective agonist of GPR35, modulating the intracellular Ca2+ release [74,75]. The expression pattern of GPR35 in the cochlea is unknown, but it is known that various immune cells, including MCs, express the GPR35 on their surface [74,76]. Here, the nonlinear type of response to cromolyn is also puzzling and requires further investigation.”

How were the ROIs chosen? Were multiple ROIs in the same image chosen and an average number calculated or one single ROI? 

For the evaluation, the area content determination method was used in ImageJ. For this purpose, a 100 µm long region between the habenula perforata and the IHC row was first selected as a "region of interest" (ROI). To analyze the area of the stained SGN in each ROI, the images were converted to a binary one. The Color Threshold function was used to highlight the selected areas in white and the non-colored background in black. Then, the area fraction of SGN in each ROI could be calculated with ImageJ.

The evaluation was performed by dividing the cochlea into 10 sections. For each section, two areas were selected that were representative of the section and were defined as ROI. Subsequently, from the 20 ROIs/per cochlea, the average ROI was calculated.

To address this point, we revised the Methods and added the following paragraph:

"For the semi-quantitative evaluation, an area of 100 µM length was selected between the habenula perforata and IHCs (ROI) and used to determine the area fraction of the SGN. The evaluation was performed by dividing the cochlea into 10 sections. For each section, two areas were selected as representative and defined as ROI. The fluorescent signal was converted into a binary image and the area fraction of the SGN in the ROI was given as a percentage."

Round 2

Reviewer 3 Report

The authors have revised the manuscript appropriately. I have no further comments.

Reviewer 4 Report

Thank you for responding to my comments - I am happy with the additional work presented and the additions to the discussion re: limitations.